# Bipartite electronic superstructures in the vortex core of $Bi_2Sr_2CaCu_2O_{8+\delta}$

T. Machida[1], Y. Kohsaka[1], K. Matsuoka[1,2], K. Iwaya[1], T. Hanaguri[1] & T. Tamegai[2]

The central issue in the physics of cuprate superconductivity is the mutual relationship among superconductivity, pseudogap and broken-spatial-symmetry states. A magnetic field $B$ suppresses superconductivity, providing an opportunity to investigate the competition among these states. Although various $B$-induced electronic superstructures have been reported, their energy, spatial and momentum-space structures are unclear. Here, we show using spectroscopic-imaging scanning tunnelling microscopy on $Bi_2Sr_2CaCu_2O_{8+\delta}$ that there are two distinct $B$-induced electronic superstructures, both being localized in the vortex core but appearing at different energies. In the low-energy range where the nodal Bogoliubov quasiparticles are well-defined, we observe the so-called vortex checkerboard that we identify as the $B$-enhanced quasiparticle interference pattern. By contrast, in the high-energy region where the pseudogap develops, the broken-spatial-symmetry patterns that pre-exist at $B = 0$ T is locally enhanced in the vortex core. This evidences the competition between superconductivity and the broken-spatial-symmetry state that is associated with the pseudogap.

[1] RIKEN Center for Emergent Matter Science, Wako, Saitama 351-0198, Japan. [2] Department of Applied Physics, The University of Tokyo, Hongo, Bunkyo-ku, Tokyo 113-8656, Japan. Correspondence and requests for materials should be addressed to T.M. (email: tadashi.machida@riken.jp) or to T.H. (email: hanaguri@riken.jp).

The electronic states in cuprates exhibit distinct features depending on energy and momentum[1,2]. The low-energy near-nodal states host the homogeneous $d$-wave superconductivity that manifests itself in the Bogoliubov quasiparticle interference (BQPI) patterns imaged by spectroscopic-imaging scanning tunnelling microscopy (SI-STM)[2–5]. BQPI is no longer observed above the doping-dependent extinction energy $\Delta_0$ and outside the diagonal line in momentum-space connecting $(\pi/a_0, 0)$ and $(0, \pi/a_0)$, where $a_0$ denotes Cu–O–Cu distance[2]. The higher-energy states near the antinode are governed by the pseudogap whose apparent magnitude $\Delta_1$ is spatially inhomogeneous and the quasiparticle excitations near $\Delta_1$ break rotational and translational symmetries of the CuO$_2$ plane[2,6–12].

To establish the relationship among these electronic states, it is indispensable to investigate how the pseudogap and the broken-spatial-symmetry state are affected when the superconductivity is suppressed. The application of $B$ is one of the ways to suppress superconductivity. It has been shown in La- and Y-based cuprates that the electronic orders that break the spatial symmetry are enhanced or even generated by $B$[13–19]. However, the detailed energy, spatial and momentum-space structures of the $B$-enhanced orders are unknown. To address this issue, we utilize SI-STM in $B$ owing to the following three advantages. First, $B$ can suppress superconductivity at the lowest temperatures where the thermal broadening effect is negligible, making it possible to study the precise energy scale of the phenomenon. Second, atomic-scale spatial resolution of SI-STM is highly beneficial not only to identify the locations of vortices but also to determine the real-space structure of vortex-induced states. Finally, by using the Fourier transformation, SI-STM acquires the momentum-space resolution even under $B$ that enables us to discuss the near-nodal and antinodal states separately.

We choose optimally doped Bi$_2$Sr$_2$CaCu$_2$O$_{8+\delta}$ (superconducting transition temperature $T_c \sim 90$ K) as a sample because of its high-quality surface necessary for SI-STM. Pioneering SI-STM studies of the vortices in Bi-based cuprates have discovered that an electronic superstructure, so-called vortex checkerboard, is nucleated in the vortex core[20–22]. Although a possible connection between the vortex checkerboard and the electronic order that breaks spatial symmetry has been discussed[20–22], the electronic states in the vortex core are still elusive, probably because the energy ranges so far studied are mostly below $\Delta_0$ and little is known about the momentum-space electronic states in the vortex core.

Our Fourier-transform SI-STM study over a wide energy range has revealed bipartite electronic superstructures in the vortex core of Bi$_2$Sr$_2$CaCu$_2$O$_{8+\delta}$. We show that the vortex checkerboard below $\Delta_0$ does not represent the electronic order but is associated with the $B$-enhanced BQPI pattern. New $B$-enhanced feature has been found at the pseudogap energy scale $\Delta_1$ where the electronic state is characterized by the broken-spatial-symmetry state[2,6–12]; the modulation amplitude of this electronic superstructure is locally enhanced in the vortex core. This consolidates the competitive relation between superconductivity and the broken-spatial-symmetry state associated with the pseudogap.

## Results

**Spectroscopic features around vortices**. Figure 1a–d show differential tunnelling conductance $g(\mathbf{r}, E, B)$ maps at energy $E = \pm 10$ meV taken at $B = 0$ and 11 T in exactly the same field of view. Here, $\mathbf{r}$ denotes the position on the surface. Vortex cores are identified as enhanced $g(\mathbf{r}, E, B)$ regions with the vortex checkerboard structure[20–22]. (Effects of $B$ on various spectroscopic quantities are shown in Supplementary Fig. 1). Figure 1e

represents the spectra spatially averaged over the regions near vortices (vortex region) and far from vortices (matrix region). (Definition of these regions are described in the 'Methods' section and Supplementary Fig. 2). As shown in Fig. 1e, the vortex alters the spectrum in two different energy regions[21–25]: the emergence of conductance humps around $|E| \sim 10$ meV $< \Delta_0$ and the suppression of the peaks at $\Delta_1$. (Detailed point spectra in a single vortex core are shown in Supplementary Fig. 3). Since these two energy regions occupy different sectors in momentum-space, near-nodal and antinodal states (Fig. 1f), it is intriguing to explore the momentum-space characters of the vortex-induced electronic states.

By taking the Fourier transformation from the spectroscopic images, we can estimate the characteristic wavevectors $\mathbf{q}(E, B)$'s of electronic-state modulations. However, in heterogeneous systems such as Bi$_2$Sr$_2$CaCu$_2$O$_{8+\delta}$, $g(\mathbf{r}, E, B)$ not only reflects the $\mathbf{r}$ dependence of the local density-of-states (LDOS) at $E$ but also includes LDOS modulations at different energies because of the $\mathbf{r}$-dependent tip elevation associated with the feedback loop. This so-called set-point effect can be suppressed by taking a ratio $Z(\mathbf{r}, E, B) = g(\mathbf{r}, +|E|, B)/g(\mathbf{r}, -|E|, B)$, which faithfully represents the ratio of the LDOS at $\pm|E|$ (refs 6,26).

**Effect of the vortex on low-energy electronic states**. First, we focus on the low-energy near-nodal region and argue the origin of the vortex checkerboard. At $B = 0$ T, the only relevant phenomenon near the node is the BQPI that is described by the octet model[4,5] in which the eight tips of the banana-shaped constant-energy contours in momentum-space dominate the quasiparticle scatterings, resulting in a set of energy-dispersive characteristic wavevectors $\mathbf{q}_i$ ($i = 1, 2, \cdots, 7$; Supplementary Fig. 4 and Supplementary Note 1). Figure 2a depicts the typical BQPI pattern seen in $Z_q(\mathbf{q}, E, B = 0$ T), the Fourier-transformed image of $Z(\mathbf{r}, E, B = 0$ T), showing the octet $\mathbf{q}_i$'s. (In the optimally doped Bi$_2$Sr$_2$CaCu$_2$O$_{8+\delta}$, signals at $\mathbf{q}_4$ and $\mathbf{q}_5$ are weak in $Z$ (refs 2,10). We have performed standard BQPI analysis (Supplementary Fig. 4 and Supplementary Note 1)[2,5], and confirm that the BQPI is restricted in the near-nodal region below the extinction energy $\Delta_0 \sim 30$ meV (Supplementary Fig. 5)[2].

An important question here is whether the vortex checkerboard, which is most prominent around $|E| \sim 10$ meV $< \Delta_0$, represents an electronic order or not. To answer this question, we have repeated the same $Z(\mathbf{r}, E, B)$ analysis at $B = 11$ T in the same field of view (Supplementary Fig. 4 and Supplementary Note 1). As shown in Fig. 2b, no additional peak is detected, whereas the intensity of each $\mathbf{q}_i$ peak depends on $B$. Figure 2c highlights the $B$-induced change obtained by subtracting $Z_q(\mathbf{q}, E, B = 0$ T) from $Z_q(\mathbf{q}, E, B = 11$ T). The enhanced intensity appears at $\mathbf{q}_1$, which represents the wavevector of the vortex checkerboard. We note that the intensities at $\mathbf{q}_2$, $\mathbf{q}_3$, $\mathbf{q}_6$ and $\mathbf{q}_7$ are suppressed by $B$ (Supplementary Fig. 5). All of these scattering $\mathbf{q}_i$'s reverse the sign of the $d$-wave superconducting gap between the initial state and the final state, while the sign is preserved in the the case of $\mathbf{q}_1$ scattering. Such suppression and enhancement of sign-reversing and sign-preserving scatterings, respectively, are exactly what are expected from the coherence factors of the quasiparticle scatterings off vortices[27–29], suggesting that the cause of the vortex checkerboard is not the electronic order but the vortex-enhanced BQPI.

The BQPI scenario is further supported by the energy dispersion in the vortex-enhanced signal at $\mathbf{q}_1$ (Fig. 2d,e). The observed $\mathbf{q}_1$ dispersion agrees well with the behaviour calculated from $\mathbf{q}_2$, $\mathbf{q}_3$, $\mathbf{q}_6$ and $\mathbf{q}_7$ based on the octet model. Altogether with the fact that the vortex-enhanced signal diminishes near $\Delta_0$ (Fig. 2f and Supplementary Fig. 5), we ascribe the vortex

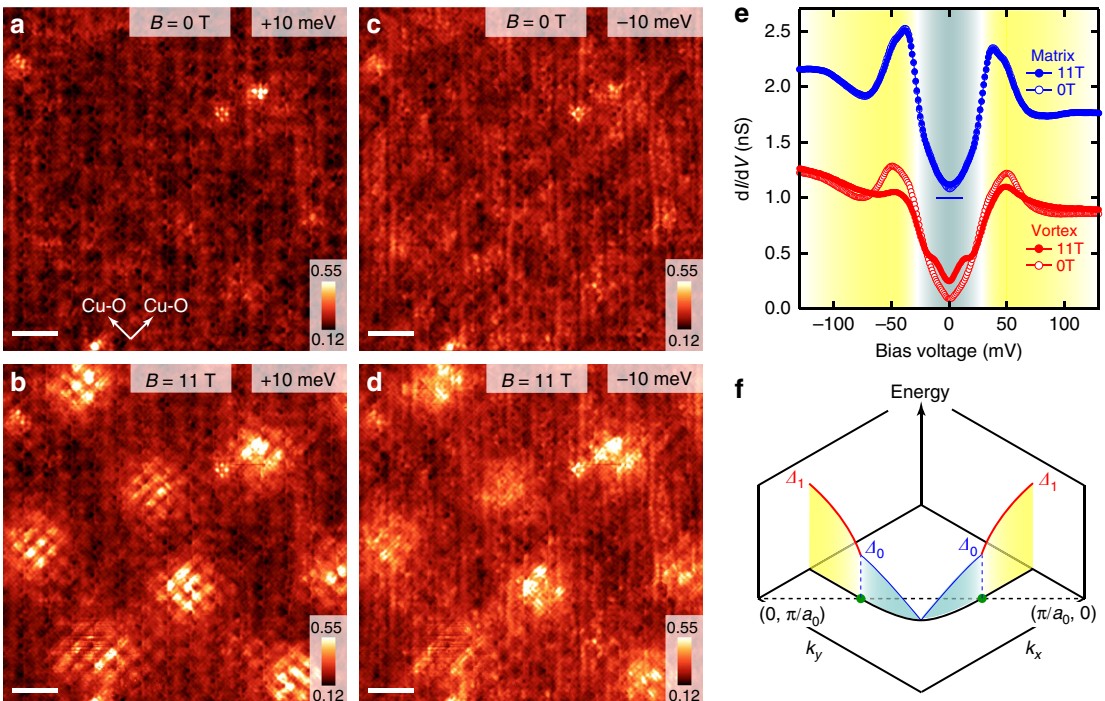

**Figure 1 | Spectroscopic features of vortices and nodal–antinodal dichotomy.** (**a,b**) Differential conductance maps at energy $E = +10$ meV in magnetic fields $B = 0$ and 11 T, respectively. White arrows in **a** denote the Cu-O bonding directions. Vortices and their internal structures (vortex checkerboard) are clearly imaged in **b**. (**c,d**) Differential conductance maps at $E = -10$ meV taken in $B = 0$ and 11 T. Scale bars, 50 Å (**a–d**), and the colour scales are in nano siemens (nS). The tunnelling conductance at each location was obtained by numerical differentiation of the current–voltage characteristics and by post-smoothing with the energy window of $\pm 2$ meV. (**e**) Comparison between tunnelling spectra taken at $B = 0$ T (open symbols) and 11 T (solid symbols). Red and blue data depict the spectra spatially averaged over the regions near vortices and far from vortices, respectively. (**f**) Schematic illustration of the excitation gap in momentum-space showing the dichotomy between the $d$-wave superconductivity near the node (light blue area) and the antinodal states governed by the pseudogap (yellow area). These two regimes are separated by the line connecting $(\pi/a_0, 0)$ and $(0, \pi/a_0)$, where $a_0$ denotes Cu–O–Cu distance[1,2].

checkerboard to the BQPI and conclude that no electronic order is nucleated in the vortex cores at $E < \Delta_0$. We note that other Friedel-type oscillations such as bound-state oscillations in the quantum-limit vortex core[22] may also be relevant (Supplementary Note 2). In any case, our observation suggests that canonical phenomenology of $d$-wave superconductivity applies in the near-nodal region even when the vortices are introduced.

**Effect of the vortex on high-energy pseudogap states.** It has been reported that the electronic feature above $\Delta_0$ is characterized by the bond-centered unidirectional electronic entity that breaks both rotational and translational symmetries[2,6–8]. For the purpose of brevity, we call this electronic entity as nanostripe hereafter. The nanostripe is reminiscent of the short-range charge order detected by X-ray scattering[30,31] and may be responsible for the weakened energy dispersion of $\mathbf{q}_1$ near $\Delta_0$ (Fig. 2d,e)[32]. Here, we investigate the effect of $B$ on the nanostripe. To visualize the nanostripe which is most prominent at $\Delta_1$, we follow the procedure used in (refs 2,7). Since $\Delta_1$ is spatially inhomogeneous, we first normalize $E$ by local $\Delta_1(\mathbf{r})$ and map $Z(\mathbf{r}, e \equiv E/\Delta_1(\mathbf{r}) = 1, B)$. Figure 3a shows $Z(\mathbf{r}, e = 1, B = 0$ T$)$ in the same field of view as Fig. 1a–d. The nanostripe in the optimally doped sample is weak in intensity[10] and is observed only in the limited regions where $Z(\mathbf{r}, e = 1, B)$ is larger. We find that these regions have larger $\Delta_1(\mathbf{r})$ and vortices tend to reside there, suggesting a vortex pinning mechanism associated with the pseudogap (Supplementary Fig. 6 and Supplementary Note 3). We have repeated the same measurement at $B = 11$ T (Fig. 3b) and have revealed that $Z(\mathbf{r}, e = 1, B)$ is enhanced in the vortex cores

(Fig. 3c). As shown in the insets of Fig. 3a–c, the structure of the nanostripe that pre-exists in the absence of the vortex core is unchanged but its contrast is enhanced. The local enhancement of nanostripe in the vortex core has also been observed in (ref. 33). We note that the local nature of the enhancement indicates that the enhancement is not caused by $B$ itself, which is almost uniform at 11 T because of the long penetration depth. Instead, it should be associated with the local suppression of superconductivity in the vortex core, representing the direct competition between superconductivity and the nanostripe.

It has been known that the nanostripe consists of two sets of wavevectors[7]: $\mathbf{Q}_{x,y} = (2\pi/a_0, 0), (0, 2\pi/a_0)$ whose inequivalent intensities represent the degree of broken rotational symmetry and $\mathbf{S}_{x,y} \sim (3/4 \times (2\pi/a_0), 0), (0, 3/4 \times (2\pi/a_0))$ that feature the broken translational symmetry. To test which broken symmetry is affected in the vortex core, we perform the Fourier analysis. By applying a mask generated from the image of vortices, we restrict our field of view in the vicinity of the vortex core to effectively extract the vortex-enhanced features (see the 'Methods' section and Supplementary Fig. 2). As shown in Fig. 3d,e, Fourier peaks corresponding to $\mathbf{Q}_{x,y}$ and $\mathbf{S}_{x,y}$ are identified at both $B = 0$ and 11 T. In the difference image shown in Fig. 3f, the intensity at $\mathbf{S}_{x,y}$ is enhanced, whereas the change at $\mathbf{Q}_{x,y}$ is small. This observation indicates that the vortex core predominantly amplifies the translational symmetry breaking that is associated with the pseudogap (Supplementary Fig. 7 and Supplementary Note 4). The vortex-enhanced nanostripe at $E \sim \Delta_1$ is reminiscent of the $B$-enhanced charge orders in Y-based cuprates[15–19], but further studies are necessary to make clear the similarities and differences between the $B$-induced charge orders in different materials.

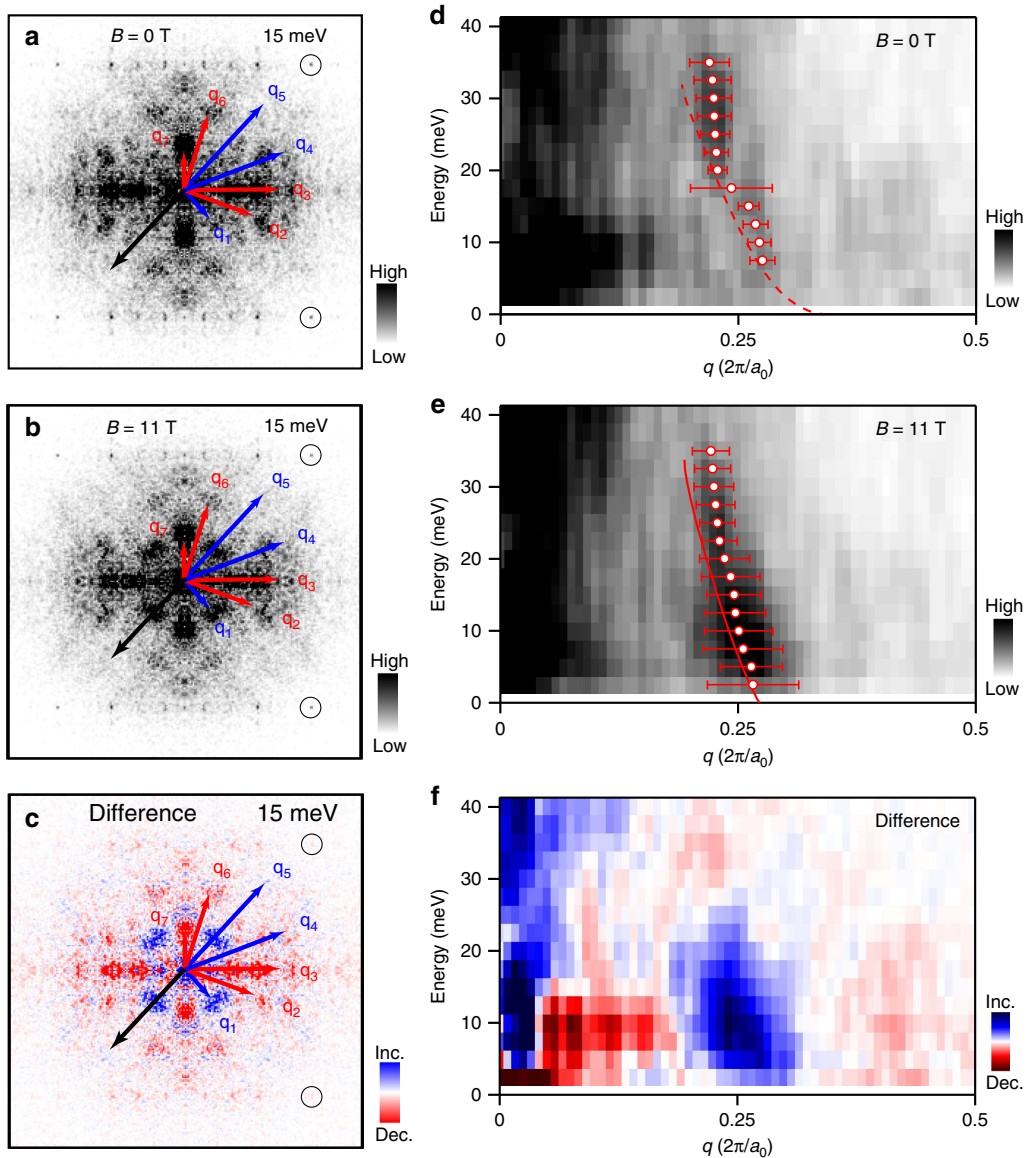

**Figure 2 | Bogoliubov quasiparticle interference origin of the vortex checkerboard.** (**a,b**) Conductance-ratio maps in scattering-vector **q** space $Z_q(\mathbf{q}, E, B)$ at energy $E = 15\,\mathrm{meV}$ in magnetic fields $B = 0$ and $11\,\mathrm{T}$, respectively. Each map was obtained by the Fourier transformation of the real-space conductance-ratio map taken with the field of view of $470 \times 470\,\text{Å}^2$, followed by the twofold symmetrization. The tunnelling conductance at each location was taken by the standard lock-in technique with a modulation amplitude of $2.5\,\mathrm{mV_{rms}}$. Red and blue arrows indicate the sign-reversing ($\mathbf{q}_2$, $\mathbf{q}_3$, $\mathbf{q}_6$, $\mathbf{q}_7$) and sign-preserving scattering ($\mathbf{q}_1$, $\mathbf{q}_4$, $\mathbf{q}_5$) wavevectors, respectively. Black circles show the Bragg spots. (**c**) Difference between a and b $Z_q(\mathbf{q}, E = 15\,\mathrm{meV}, B = 11\,\mathrm{T}) - Z_q(\mathbf{q}, E = 15\,\mathrm{meV}, B = 0\,\mathrm{T})$. (**d,e**) Energy-dependent line profiles taken along the black arrows in **a** and **b**, respectively. Red open circles denote the position of the $\mathbf{q}_1$ peak that has been determined by fitting the line profile at each energy by a Lorentzian function. Error bars indicate the full width at half maximum of the fitted peak. Dashed red line in **d** and solid red line in **e** are the dispersions calculated from $\mathbf{q}_2$, $\mathbf{q}_3$, $\mathbf{q}_6$ and $\mathbf{q}_7$ based on the octet model. Note that the $B$-induced signal exhibits the dispersion that is consistent with the Bogoliubov quasiparticle interference. (**f**) Energy-dependent line profile taken along the black arrow in **c**, showing the $B$-induced change.

## Discussion

It is noteworthy to argue the spectroscopic features in the vortex core where the spectral weights at $|E| \sim \Delta_1$ and $|E| \sim 10\,\mathrm{meV}$ are suppressed and enhanced, respectively (Fig. 1e). Given the close correlation between the pseudogap and the nanostripe (Supplementary Fig. 7 and Supplementary Note 4)[2,6–11], the missing weight at the pseudogap energy $\Delta_1$ along with the enhanced nanostripe demands careful consideration. It may be possible that the enhanced nanostripe causes the extra quasiparticle decoherence[34] that suppresses the weight at $|E| \sim \Delta_1$. However, this scenario can not account for the new states created at $|E| \sim 10\,\mathrm{meV}$. Here, we point out that the

observed spectral weight transfer from $|E| \sim \Delta_1$ to $|E| \sim 10\,\mathrm{meV}$ can naturally be explained if the superconducting gap is as large as $\Delta_1$; the missing weight at $|E| \sim \Delta_1$ is originated from the suppression of superconductivity, being transferred to the vortex-bound states at $|E| \sim 10\,\mathrm{meV}$ (refs 22,35). Since two different orders, in the present case, superconductivity and the nanostripe, can not share the same region in energy-momentum-space, it is plausible that they are nearly degenerated but competing states near the antinode[15,19,35,36].

This scenario provides an insight into the heterogeneous electronic states in real-space. Randomly-dispersed dopants and defects locally perturb the balance between superconductivity and

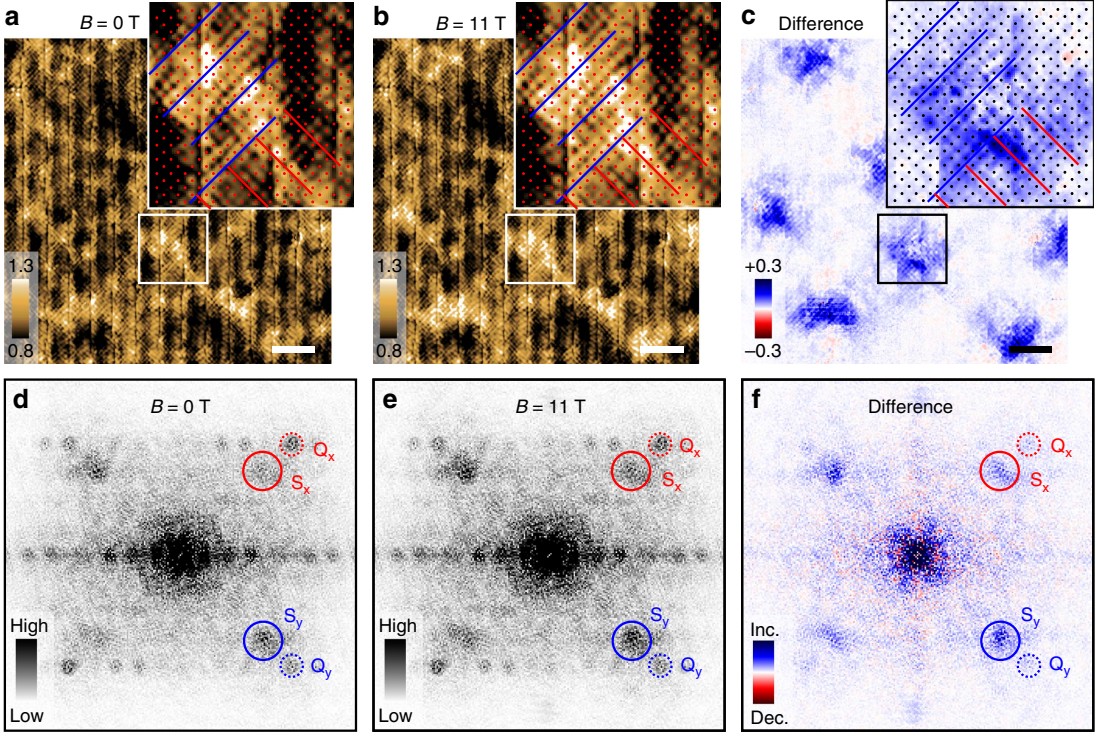

**Figure 3 | Magnetic field effect on the nanostripe at the pseudogap energy scale.** (**a,b**) Conductance-ratio maps $Z(\mathbf{r}, e=1, B)$ at the position ($\mathbf{r}$)-dependent pseudogap energy $\Delta_1$ in magnetic fields $B=0$ and 11 T, respectively. Here, $e \equiv E/\Delta_1$ where $E$ denotes energy. The tunnelling conductance was obtained by numerical differentiation of the current–voltage characteristics and by post-smoothing with the energy window of $\pm 10$ meV. (**c**) Difference between (**a,b**) $Z(\mathbf{r}, e=1, B=11\,\text{T}) - Z(\mathbf{r}, e=1, B=0\,\text{T})$. Scale bars, 50 Å. Insets of **a–c** are the magnified images of the regions marked by boxes in the main figures. Red and blue lines in the insets indicate the directions of the nanostripe. Dots denote the locations of the Cu atoms. (**d,e**) Fourier-transformed images of **a** and **b**, respectively. The field of views are restricted in the vicinity of the vortices by applying a mask. (**f**) Difference between **d** and **e**. In these Fourier-transformed images, dotted and solid circles indicate the characteristic wavevectors of the nanostripe $\mathbf{Q}_{x,y} = (2\pi/a_0, 0), (0, 2\pi/a_0)$ and $\mathbf{S}_{x,y} \sim 3/4 \times (2\pi/a_0, 0)$, $3/4 \times (0, 2\pi/a_0)$, respectively.

the nanostripe, and consequently bring about the nano-scale mixture of them. The observed vortex-enhanced nanostripe suggests that the vortex core is one of these perturbations that affects the competition near the antinode. This competition scenario of the $B$-enhanced electronic order is based on our unambiguous identification of the energy scale of the phenomenon and can be further tested by controlling the energy scales of superconductivity and the pseudogap by doping.

## Methods

**SI-STM measurements and sample preparation.** SI-STM experiments were performed at a temperature of 4.6 K with a modified commercial low-temperature ultra-high-vacuum STM (Unisoku USM-1300) installed in RIKEN[37]. Single crystals of $Bi_2Sr_2CaCu_2O_{8+\delta}$ were grown at the University of Tokyo by the traveling-solvent floating-zone method, and were annealed to have optimal hole concentration. The superconducting transition temperature $T_c \sim 90$ K was determined by the magnetization measurement. The sample was cleaved *in situ* at $\sim 77$ K to obtain a clean and flat (001) surface and was transferred quickly to the STM unit kept at 4.6 K. We used an electro-chemically etched tungsten wire as an STM tip, which was cleaned and characterized *in situ* with a field-ion microscope. All the SI-STM data were taken with the feedback set point at a sample bias voltage of $-150$ mV and a tunnelling current of 150 pA. $B$ was applied perpendicular to the cleaved (001) surface. Whenever we changed $B$, the sample was heated up to $\sim 30$ K to make the vortex distribution inside the sample uniform.

**Procedure to correct image distortions.** It is well-known that electronic states of $Bi_2Sr_2CaCu_2O_{8+\delta}$ are heterogeneous especially near $\Delta_1$. Therefore, to argue the vortex-induced local changes in the electronic state, it is indispensable to compare two SI-STM data sets, with and without a magnetic field, in exactly the same field of view. This is challenging because the actual SI-STM images are inevitably distorted in an uncontrollable manner due to the creeping of the piezoelectric scanner and so on. To obtain distortion-free images from the observed ones, we utilize the so-called Lawler–Fujita algorithm[7], which estimates the local distortion

as a phase shift in the crystal-lattice modulations. We first estimate and correct the distortions in the topographic images simultaneously taken with the $g(\mathbf{r}, E, B)$ images and correct the spectroscopic images, $g(\mathbf{r}, E, B)$ and $Z(\mathbf{r}, E, B)$, using the same local distortions. Various images obtained by SI-STM after the correction are shown in Supplementary Fig. 1.

**Definition of the vortex and matrix regions.** Vortices are most clearly seen in the difference map $\delta g(\mathbf{r}, E=+10\,\text{meV}, B=11\,\text{T}) \equiv g(\mathbf{r}, E=+10\,\text{meV}, B=11\,\text{T}) - g(\mathbf{r}, E=+10\,\text{meV}, B=0\,\text{T})$ (Supplementary Fig. 2a). We first apply low-pass filter to the $\delta g(\mathbf{r}, E=+10\,\text{meV}, B=11\,\text{T})$ map with a cutoff wavelength of $q \sim 0.05 \times 2\pi/a_0$. Constant contours of this filtered image are used for the boundaries of the masks (Supplementary Fig. 2b). The vortex and matrix regions used to examine the vortex-induced change are shown in Supplementary Fig. 2c. The spectra shown in Fig. 1e are the spectra averaged in these regions. In Supplementary Fig. 3, we depict the detailed point spectra near the vortex before averaging. The mask used for the Fourier analyses (Fig. 3d–f) and the restricted $Z(\mathbf{r}, E, B)$ maps at $B=0$ and 11 T are indicated in Supplementary Fig. 2d–f, respectively.

**Data availability.** All relevant data are available on request, which should be addressed to T.M. or T.H.

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

## Acknowledgements

We thank J.C. Davis, A.N. Pasupathy and S. Tajima for discussions and comments. This work was partly supported by Grant-in-Aid for Scientific Research from the Ministry of Education, Culture, Sports, Science and Technology of Japan (Grant No. 20244060).

## Author contributions

T.M. carried out the experiments and the data analyses with assistance from Y.K., K.M., K.I., and T.H. T.T. grew the single crystals. T.H. supervised the project. T.M. and T.H. wrote the manuscript.

## Additional information

**Competing financial interests:** The authors declare no competing financial interests.

