## [Peer Review File · Nature Communications]

Reviewers' comments:

Reviewer #1 (Remarks to the Author):

I am much happier with the new paragraph discussing the interplay between the B enhanced QPI at 10-15 meV and the B-suppressed spectral weight at the pseudogap energy. I bet the authors are fully aware of the fact that the divergence of the pseudogap temperature and the "charge order temperature" in underdoped regime poses challenge on the identification of the pseudogap with the gap created by charge ordering. This is a question that the entire STM community studying cuprates needs to face in the future.

I have no more objections to the publication of this paper.

Reviewer #2 (Remarks to the Author):

As written in my original reports, this is an interesting magnetic field-dependent STM study of the electronic structure and energy scales found in optimally doped Bi2212. I previously suggested Nature Communications would be an appropriate journal for this work and retain this opinion. With except for a single suggestion (given below), I recommend to publish this manuscript as it is.

Suggestion:

As the authors suggest a connection between nano-stripe order and pseudo gap physics, it would be fair to cite Nature 463, 519 (2010) which reported a connection between a broken rotational order and the pseudo gap phase in YBCO.

Reply to the Reviewer #1:

We thank the reviewer for recognizing the value of the manuscript. We agree with the reviewer that there still remain issues that should be addressed in the future. As we wrote in the end of the text, the doping dependence of the B-enhanced order may give us further insights.

Reply to the Reviewer #2:

We thank the reviewer for the recommendation of the publication in Nature Communications and the suggestion of the reference. We cite it in the revised manuscript (Ref. 12).